# Electrically Small Wideband Monopole Antenna Partially Loaded with Low Loss Magneto-Dielectric Material

Aladdin Kabalan, Ala Sharaiha * and Anne-Claude Tarot

Institut d'Electronique et des Technologies du numéRique (IETR), University of Rennes 1, 263 Avenue du Général Leclerc, 35700 Rennes, France; aladdin-kabalan@hotmail.com (A.K.); anne-claude.tarot@univ-rennes1.fr (A.-C.T.)
* Correspondence: ala.sharaiha@univ-rennes1.fr

**Abstract:** A miniaturized new topology of the planar monopole antenna using a Magneto-Dielectric Material (MDM) is proposed in this paper. The antenna element is realized by introducing slots partially covered by the MDM. We optimized and modified the MDM topology and dimensions to enhance the impact of this material on the planar monopole antenna, including slots in its structure. This new monopole shows a miniaturization rate of 60% of the antenna's height (51 cm antenna's height is miniaturized to 20 cm) by covering only 5% of the antenna surface by the MDM. The measured results show the antenna's central working frequency of 130 MHz, while the bandwidth is 30% using a broadband matching circuit using the Real Frequency Technique (RFT).

**Keywords:** magneto-dielectric materials; planar monopole antenna; miniaturization

## 1. Introduction

Planar monopole antennas are widely used in many communication systems due to their simple structure, compact size, ease of manufacture and implementation, especially in aeronautical applications [1–4] in the VHF and UHF frequency bands. The antenna size is relatively important in these bands, which causes an integration problem for most of the applications in these bands. Therefore, a small antenna size is necessary and becomes a critical problem in communication systems.

Among the various existing miniaturization techniques, the use of semi-massive Magneto-Dielectric Materials (MDM) [5] is one of the most promising for reducing the antenna size. Instead of using a material with high permittivity, the same miniaturization can be obtained by using MDM of the same refractive index $n = (\mu_r \varepsilon_r)^{0.5}$ with moderate permittivity and permeability avoiding highly concentrated field confinement, and the medium is far less capacitive when compared to the dielectric-only high permittivity, which results in low antenna efficiency and narrow-band operation [6,7]. On the other hand, antenna matching is difficult in a medium of high permittivity because of the high-quality factor value.

Since the work of Hansen and Burke [8], magneto-dielectric materials have been known to be more advantageous in antenna miniaturization and, in ultra-wideband antenna applications, since the work of Volakis [9]. Many other papers have confirmed, at least in theory, that magneto-dielectric materials give larger bandwidths and also better efficiencies compared to dielectric ones [8,10–12]. Furthermore, the MDM should have sufficiently low dielectric and magnetic losses tangents to ensure good performance of antennas made of them. Recent papers have proposed a new hexaferrite structure [13] with a high FOM (Figure of merit of the product $\mu'$, Q and the operating frequency f) which has a strong potential for low-loss high-frequency applications.

These last years, modern material manufacturing technology has made it possible to design composite or nanocomposite substrates with different kinds of magnetic inclusions mixed with dielectric host materials. Most of these MDM substrates were used to miniaturize planar antennas (patch, PIFA and IFA), with the ratio $\mu_r/\varepsilon_r$ of 1.5 [12,14], 5.9 [15]

and 2.1 [16], and loss tangents (tan $\delta_{\varepsilon,\mu}$) less than 0.1 for a frequency band up to 700 MHz. Showing a better performance compared to commercial dielectric materials.

In [17], we studied the effect of MDM composed of NiZnCo ferrite, which in the VHF band has a higher permeability ($\mu_r = 16$) than the permittivity ($\varepsilon_r = 12$), on the miniaturization of a planar monopole antenna in the VHF band 100–200 MHz. We achieved an optimum miniaturization rate of 15%, covering only 12% of the total antenna size where the intensity of the surface current is maximum. In order to increase the magnetic impact of the MDM, we propose, in this paper, modifying the monopole topology by introducing slots and by optimizing the size and the position of the MDM, as well as its geometry. Then the effect of the air gap, due to the sheet metal thickness when using Planar MDM (P-MDM), on the magnetic field distribution was investigated.

The paper is organized in the following way: after briefly presenting the fabricated MDM and the reference antenna in Section 2, we presented the principal results of the antenna miniaturization using a small portion of P-MDM and introduced slots in Section 4. Then in Section 5, we presented a parametrical study, where the antenna geometry and the dimensions of the P-MDM were optimized, and then in Section 6, a first prototype using P-MDM was designed and validated by measurements. Finally, Section 7 presents some concluding remarks.

## 2. MDM Description

An MDM was developed with the objective of obtaining a high permeability with a low loss in the frequency band of 100–200 MHz. The MDM based on a ferrite Ni-Zn-Co was elaborated by the Lab-STICC laboratory, and the method is presented in [5].

The relative permeability ($\mu_r$) and permittivity ($\varepsilon_r$) spectrum, as well as the magnetic and dielectric loss tangent in the frequency band of 100–200 MHz, are shown in Figure 1a,b, respectively.

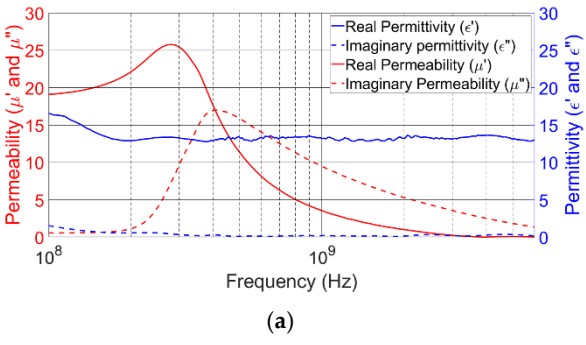 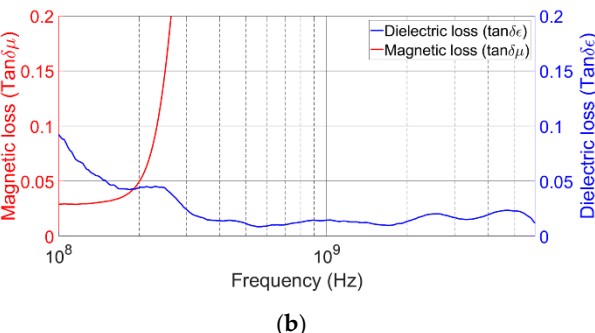

(**a**)                                                                           (**b**)

**Figure 1.** Manufactured MDM spectrum in the band 10 MHz–6 GHz. (**a**) Typical measured relative permeability and permittivity. (**b**) Magnetic and dielectric loss tangents.

It was observed that between 100 and 200 MHz, the ratio $\mu'/\varepsilon'$ is greater than unity and varies between 1.46 and 1.37. Magnetic losses are low, between 0.03 and 0.05. This property is related to the nanometric dimensions of grains that are preserved during heat treatment.

## 3. Geometry of the Planar Monopole Antenna

An ultra-wideband (UWB) planar monopole antenna is designed to operate in the VHF band (118–156 MHz), with the dimensions shown in Figure 2a (antenna A1); the other dimensions W2 = 5 mm and θ = 40° are optimized for a larger bandwidth. Figure 2b shows the antenna input impedance with a bandwidth of 31.5% (for VSWR < 2.5) around its resonance frequency $f_0$ = 130 MHz. We assumed in our study that the ground plane of the monopole antenna has infinite dimensions.

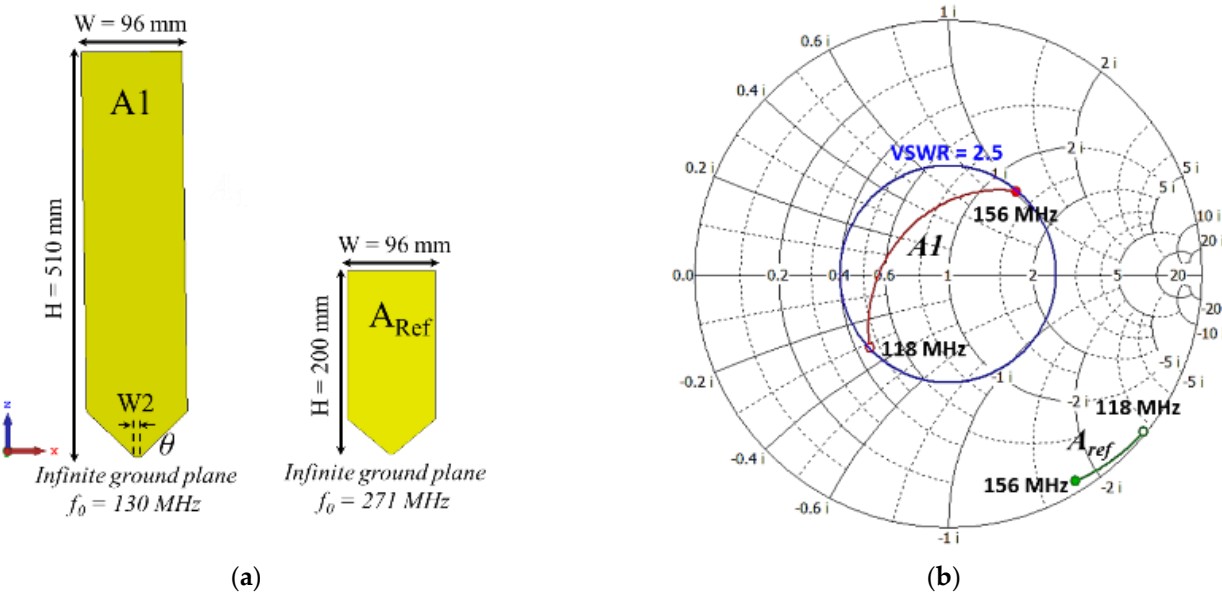

**Figure 2.** (**a**) Simple monopole antenna A1 of the height of 510 mm, and the reference monopole antenna $A_{Ref}$ of the height of 200 mm. (**b**) Smith chart of the antenna's impedances in the band 118 to 156 MHz.

The planar monopole antenna was chosen for our project of aircraft application, which is the suitable type of antenna for the VHF band. The scope statement goal of reducing the antenna height from 51 cm to 20 cm corresponds to a miniaturization rate of 60.7%. The monopole antenna with 20 cm of height called "$A_{Ref}$" is shown in Figure 2a, and its input impedance is represented on the Smith chart in Figure 2b. The size of the antenna A1 is in the order of $\lambda/4.5$ at the frequency 130 MHz, while the reference antenna $A_{ref}$ must be miniaturized up to $\lambda/11.5$.

## 4. Miniaturization of the Monopole Antenna Using a P-MDM

In this section, we study the effect of the Planar MDM (P-MDM) to shift the resonance frequency of the reference antenna $A_{Ref}$, considering for the moment that the metal thickness of the antenna $t_a$ is zero.

We showed in previous work [17] that it is sufficient to cover the regions close to the source where the intensity of the magnetic field is the highest to obtain an optimum miniaturization rate. Therefore, by covering the reference antenna partially with the P-MDM (3 mm thick, 60 mm height) on both sides (see Figure 3a), antenna A2 is obtained. The latter resonates at 176 MHz lowering the frequency by only about 35%. We should emphasize that our objective was to attain 130 MHz without increasing the size of the antenna. The P-MDM thickness is limited to 3 mm because of technical reasons, knowing that the material thickness has a direct effect on the effective medium (effective permeability and permittivity). In our application, the increase in the P-MDM thickness leads to an effective permeability more important. However, the objective is to have the desired miniaturization effect of the P-MDM with the minimum quantity of the material.

In order to achieve high miniaturization rates, we increased the impact of the P-MDM. For this, we proposed the antenna A3 structure with the inclusion of two open-ended slots in this area (see Figure 3b). These two slots, with a length of 50 mm and a thickness of 5 mm, serve to concentrate the surface currents in the area of intersection between them (Figure 3c), which consequently leads to an increase in the magnetic field intensity in the zone covered by the P-MDM. Thus, an increase in the effective permeability of the medium is obtained, providing a higher miniaturization rate.

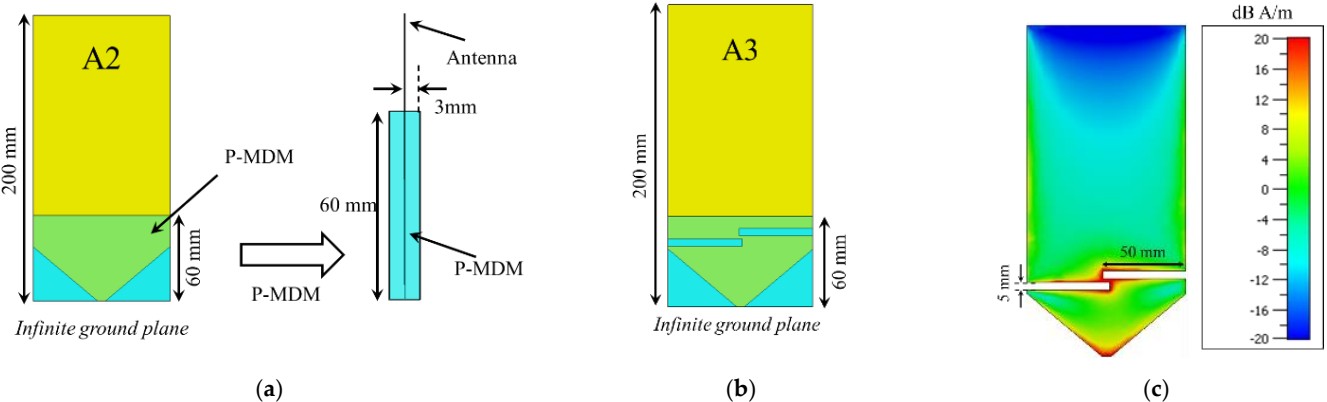

**Figure 3.** Monopole antenna miniaturized by the P−MDM: (**a**) without slots (A2) and (**b**) with two slots of 50 × 5 mm (A3). (**c**) Density of the surface currents (A/m), which corresponds to the intensity of the magnetic field.

The effective medium can be calculated numerically by simulation, by determining the shift of the resonance frequency caused by the material, using the following equation:

$$f_{0,\text{MDM}} = \frac{f_{0,air}}{\sqrt{\varepsilon_{eff}\mu_{eff}}} \tag{1}$$

where $f_{0,air}$ and $f_{0,\text{MDM}}$ are the resonance frequencies of the antenna without and with MDM, respectively. $\varepsilon_{eff}\mu_{eff}$ are the effective permittivity and permeability, respectively. In the regions close to the source, the intensity of the electric field is the lowest; therefore, the permittivity of P-MDM does not have any effect on the resonant frequency of the antenna, and the effective permittivity is neglected.

For a $\mu_r$ of about 20, the effective permeability increases from about 2.28 for the antenna without slots A2 to 4.6 for the antenna with the two slots A3, respectively (see Figure 4). Consequently, the resonance frequency decreases from 175 MHz to 130 MHz, increasing the miniaturization rate to 60.7% (see Figure 5).

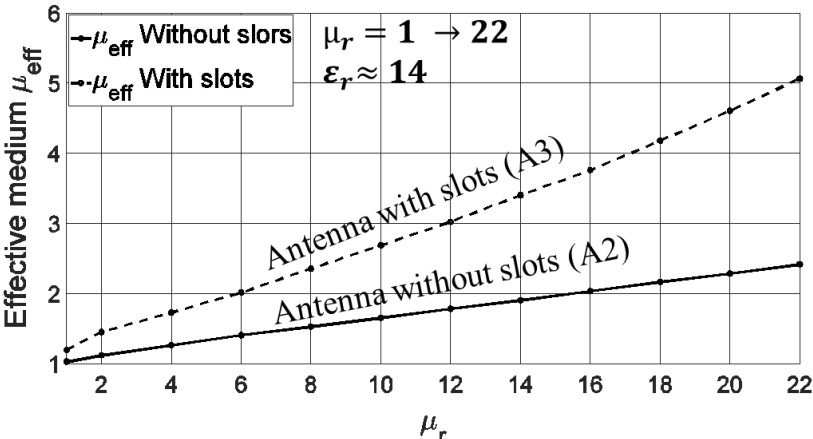

**Figure 4.** Effective medium in function of the relative permeability for both miniaturized antennas: without slots (A2) and with two slots (A3).

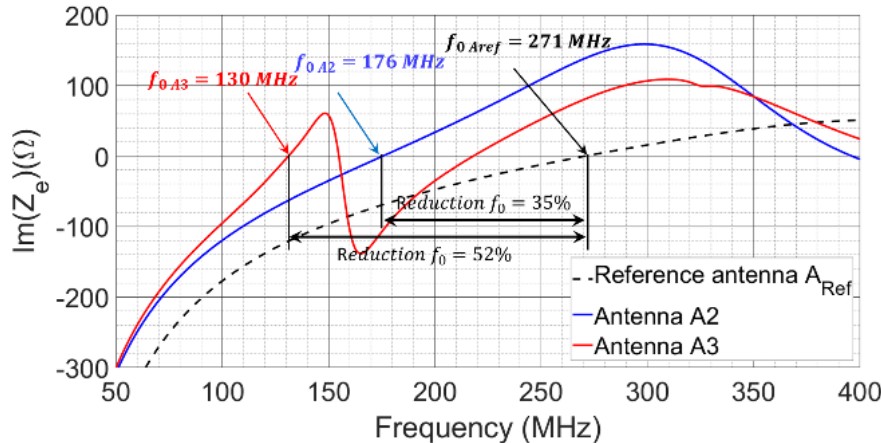

**Figure 5.** Imaginary parts of the input impedance of 3 antennas: $A_{Ref}$, A2 and A3.

## 5. Parametrical Study

In this section, we studied the impact of the size of the P-MDM used ($W_{MDM}$ and $H_{MDM}$) as well as the slot length $L_{slot}$ and height $H_{slot}$ on the resonance frequency $f_0$ and the corresponding $\mu_{eff}$ (see Figure 6).

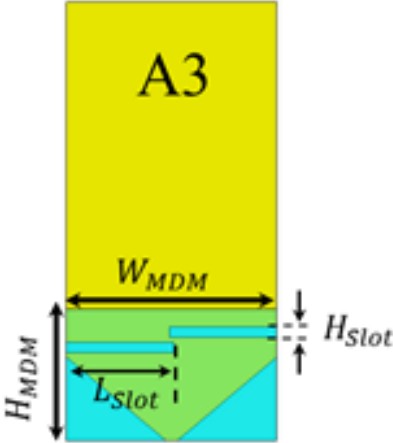

**Figure 6.** Parametrical study of the Antenna A3 with the parameters: $H_{MDM}$, $W_{MDM}$, $L_{slot}$ and $H_{slot}$.

The initial parameters are the following: $W_{MDM}$ = 96 mm, $h_{MDM}$ = 20 mm, $L_{slot}$ = 50 mm, $H_{slot}$ = 5 mm.

### 5.1. Optimization of the Position and the Dimensions of the P-MDM

Here, we considered $H_{MDM}$ as the height of the P-MDM. Several values of the $H_{MDM}$ were tested between 20 mm and 55 mm when the other dimensions were fixed with $L_{slot}$ = 50 mm, $H_{slot}$ = 5 mm and $W_{MDM}$ = 96 mm (Figure 7a). It is shown in Figure 7b that the reduction rate of $f_0$ as a function of $H_{MDM}$ decreased only by about 2%, corresponding to a $\Delta\mu_{eff}$ of 0.35.

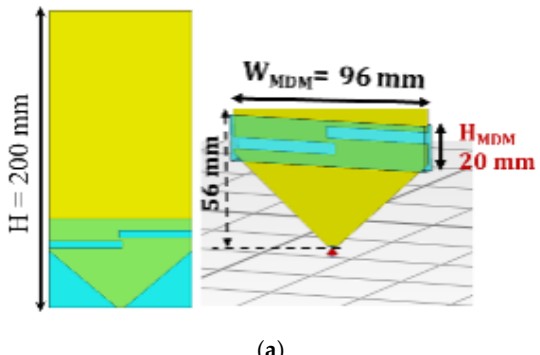

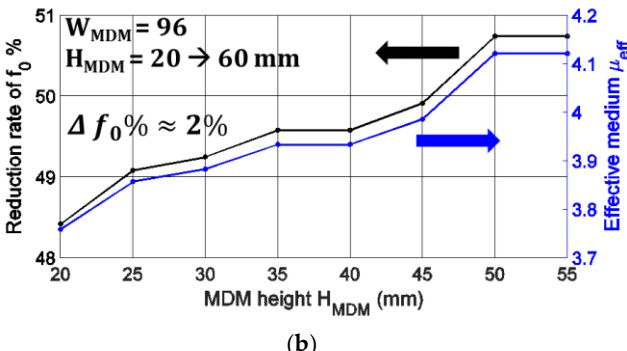

(a)

(b)

**Figure 7.** (**a**) Monopole antenna of the height 200 mm with two slots of the width 50 mm. (**b**) Reduction rate of $f_0$ and $\mu_{eff}$ in function of P-MDM height $H_{MDM}$ with a constant $W_{MDM}$ = 96 mm.

Next, we varied the width $W_{MDM}$ from 8 mm to 96 mm with a constant height of $H_{MDM}$ = 20 mm (Figure 8a) and keeping $L_{slot}$ and $H_{slot}$ the same. We can see in Figure 8b that at $W_{MDM}$ = 40 mm, the reduction rate of $f_0$ is equal to 41% and converges slowly to 49% for the maximum width $W_{MDM}$ = 96 mm.

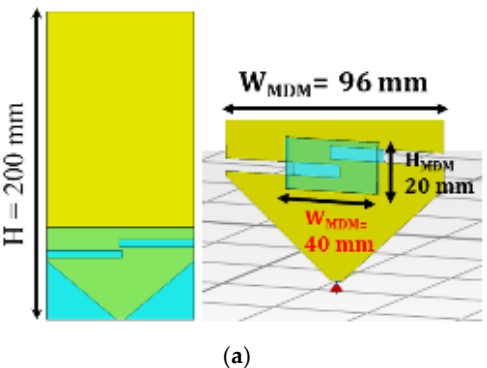

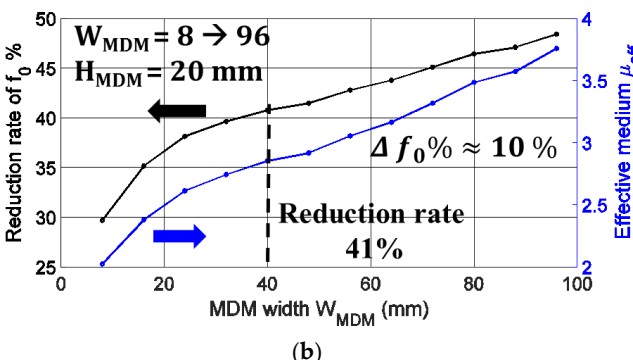

(a)

(b)

**Figure 8.** (**a**) Monopole antenna of the height 200 mm with two slots of the width 50 mm. (**b**) Reduction rate of $f_0$ and $\mu_{eff}$ in function of P-MDM width $W_{MDM}$ with a constant $H_{MDM}$ = 20 mm.

### 5.2. Slots Dimensions Effect on the Miniaturization Rate

Here, $L_{slot}$ varies between 10 mm and 90 mm (Figure 9a) with a constant $H_{slot}$ of 5 mm. The P-MDM size is fixed to $W_{MDM}$ = 4 cm and $H_{MDM}$ = 2 cm. We noticed that the reduction rate increases slowly until the slot length of 40 mm, where the intersection between the two slots begins. For $L_{slot}$ > 40 mm, the effective medium increased quickly to reach the value of 10.4, corresponding to a 69% size reduction for $L_{slot}$ = 70 mm. We noticed that the reduction rate increases more slowly for $L_{slot}$ > 70, where the slots exceed the P-MDM border.

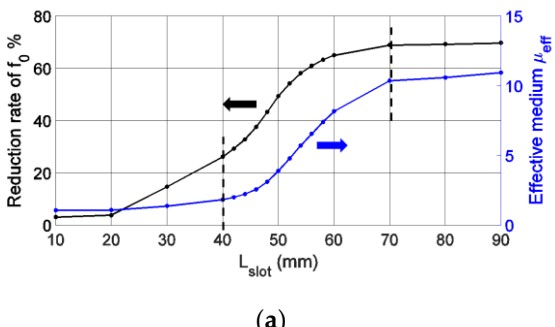

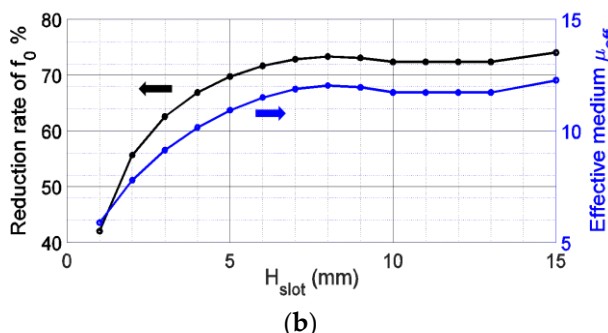

(a)

(b)

**Figure 9.** (**a**) Reduction rate of $f_0$ and $\mu_{eff}$ in function of the slot's length $L_{slot}$. (**b**) Reduction rate of $f_0$ and $\mu_{eff}$ in function of the slot's height $H_{slot}$.

Therefore, we obtained high miniaturization rates by increasing $L_{slot}$ more than 40 mm. This is because we increased the surface area of the intersection between the two slots, and therefore the intensity of the magnetic field in the P-MDM was increased, which allows for modifying the effective medium. We found a similar effect when we varied the $H_{slot}$ between 1 and 10 mm with a constant $L_{slot}$ of 90 mm (Figure 9b), the $\mu_{eff}$ increases until the slot height of 10 mm, where the slots exceed the P-MDM.

Regarding these results, we suggest reducing the size of the P-MDM to 4 cm × 2 cm to cover only the region of the intersection between the two slots. We obtained the antenna A4 (see Figure 10a). In the realized antenna, we had a problem with an air gap of about 1 mm between the MDM and the antenna surface. This air gap can reduce the intensity of the magnetic field inside the MDM, and as a consequence, that reduces the miniaturization rate. Therefore, to compensate for the effect of the air gap on the miniaturization rate, we had to adjust the dimensions of the slot to increase the intensity of the magnetic field inside the MDM. In order to have the resonance at 130 MHz, we had to increase the slot dimensions to $L_{slot}$ = 93 mm (instead of 50 mm) and $H_{slot}$ = 10 mm (instead of 5 mm), which increased the $\mu_{eff}$ value to 4.34. This gave us the antenna A5 with a resonant frequency of 130 MHz (see Figure 10b).

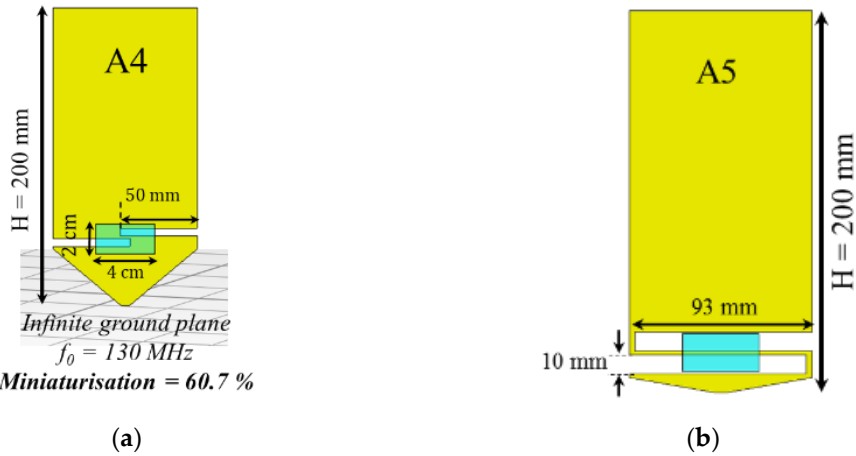

(a) (b)

**Figure 10.** (**a**) Miniaturized monopole antenna A4 by the P-MDM of dimensions 4 × 2 cm with two slots of length 51.3 mm. (**b**) Miniaturized monopole antenna A5 by the P-MDM with two slots of dimensions 93 × 10 mm.

## 6. Realization and Experimental Validation

The antenna A5 was realized with 1 mm thick brass plate and measured using a 1-m diameter ground plane. The two P-MDM of the antenna A5 were integrated into two plastic supports to tighten the plates as much as possible on the antenna (see Figure 11).

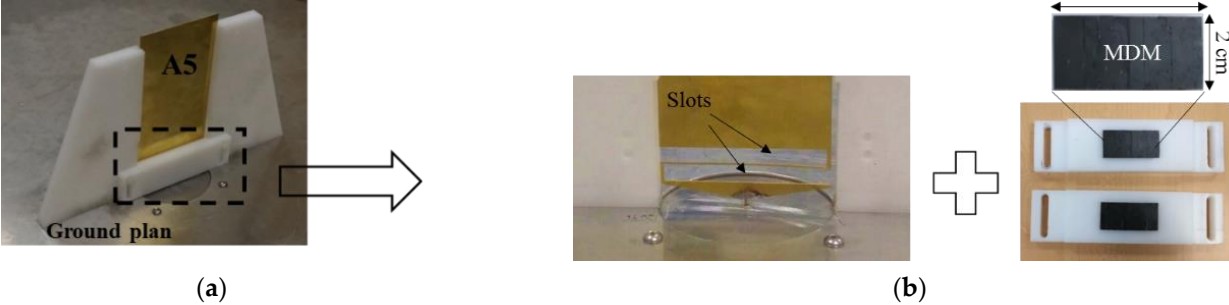

(a) (b)

**Figure 11.** (**a**) planar monopole A5 realized with two P-MDM. (**b**) Slots and two P-MDM.

Figure 12 shows the imaginary and real parts of the measured and simulated impedance of the antenna A5. There is a good agreement between the results in the band 118–156 MHz.

The measured antenna resonates almost at the same resonance frequency found in the simulation, which is around 130 MHz.

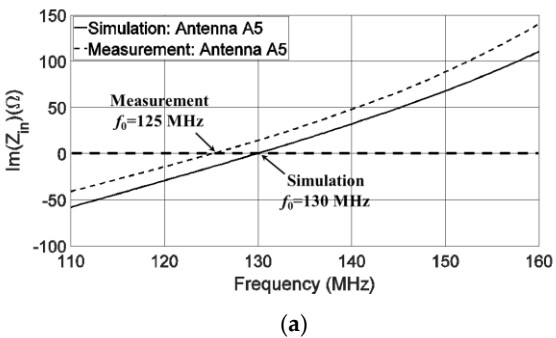 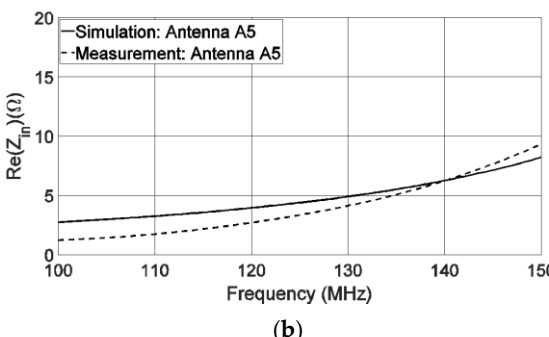

(**a**)                       (**b**)

**Figure 12.** Imaginary (**a**) and real (**b**) parts of the measured and simulated input impedance of the antenna A5.

Therefore, a 60.7% of size reduction can be obtained using P-MDM. However, the use of a matching circuit is required to tune the input reactance to zero. We matched the antenna's input impedance to cover 118–156 MHz in the VHF band for VSWR less than 3.5 (S11 < −5 dB). In order to achieve this goal, we proposed using the « Real Frequency Technique (RFT) [18] to design the broadband matching circuits shown in Figures 13 and 14. The matching circuit of the antenna A5 was made on an FR4 substrate of a thickness of 1.6 mm and using SMD components of the Murata series [19].

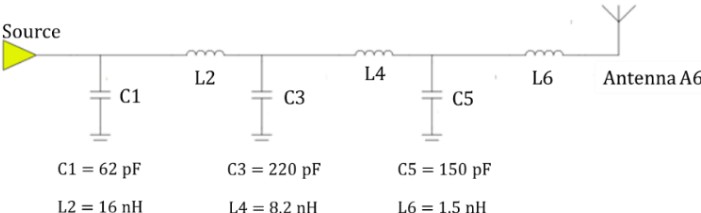

**Figure 13.** Matching circuit of the miniaturized monopole antenna A5 for the band 118–156 MHz.

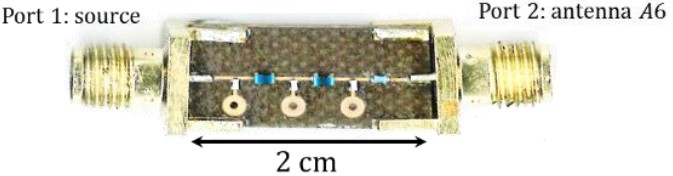

**Figure 14.** Realized matching circuit of the antenna A5.

Then, by connecting the antenna to port 2 of the matching circuit, the reflection coefficient S11 was measured at the circuit input and compared with the simulation result in Figure 15. As the individual component tolerances have a profound effect on the overall circuit performance, the shaded curves show the potential errors of the reflection coefficient S11 due to the component tolerances. Therefore, a smaller error between measured and simulated results can be obtained using components with smaller tolerance values. In addition, the realized gain of the matched antenna A5 was validated by the measurements with a good agreement between the results, as shown in Figure 16. This gained respect for the scope statement goals of our project. The simulated radiation pattern of the matched antenna A5 is shown in Figure 17. We noticed that the highly miniaturized monopole antenna A5 maintains its omnidirectional radiation performance well.

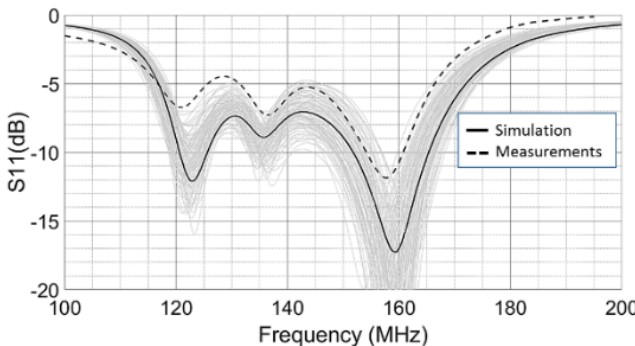

**Figure 15.** Measured and simulated reflection coefficient S11 of the matched antenna A5 including the sim-ulated individual component tolerances effects..

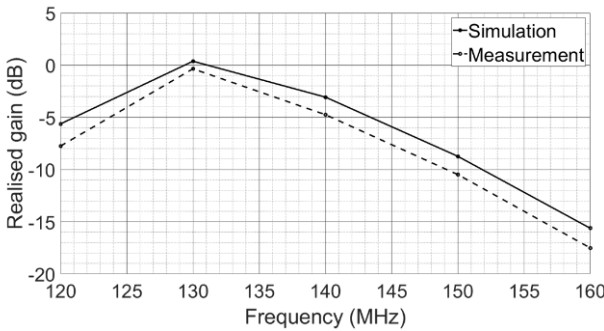

**Figure 16.** Measured and simulated realized gain of the miniaturized and matched antenna A5.

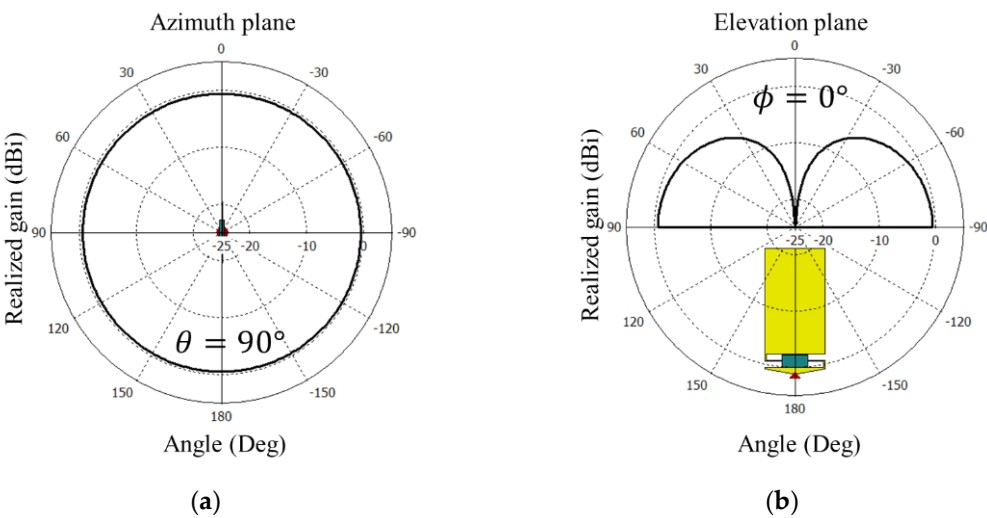

(**a**)                    (**b**)

**Figure 17.** Simulated radiation pattern of the miniaturized and matched antenna A5 at the frequency 130 MHz. (**a**) in azimuth plane (**b**) in elevation plane.

## 7. Conclusions

In this paper, we proposed a new geometry of planar monopole antenna using slots to take advantage of the low loss Planar Magneto-Dielectric Material (P-MDM) in order to achieve very high miniaturization rates. By placing the slots between two small pieces of the P-MDM, we increased the concentration of the magnetic field intensity and consequently obtained a higher effective magnetic medium and then a higher miniaturization rate. The proposed miniaturization method was validated by the measurements of the miniaturized antennas shown in this article.

**Author Contributions:** Supervision, A.S.; Writing—original draft, A.K., A.S. and A.-C.T. All authors have read and agreed to the published version of the manuscript.

**Funding:** This work was carried out in the framework of the MISTRAL project, which is funded by the French National Research Agency (ANR) under contract number ANR−15−CE24−0030.

**Conflicts of Interest:** The authors declare no conflict of interest.

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
