# Peer review of "Electrically Small Wideband Monopole Antenna Partially Loaded with Low Loss Magneto-Dielectric Material"

_2673-8724, doi:10.3390/magnetism2030017_

Round 1

Reviewer 1 Report

In this paper, the authors achieved the miniaturization of a 130-MHz planar monopole antenna using a low-loss planar magneto-dielectric material. The new antenna topology was realized by introducing slots partially covered by the planar magneto-dielectric material and the miniaturization rate was about 60%. While the paper is quite straightforward, it could be further improved by the following suggestion or discussion.

1. In Introduction, the review on the magneto-dielectric materials is insufficient. Discussion on different types of the MDMs and their capabilities on antenna miniaturization should be mentioned. The discussion in the paper shown below is closely related to this topic and is worth citing.

Q. Li, Y. Chen, C. Yu, L, Young, J. Spector, V.G. Harris, Emerging magnetodielectric materials for 5G communications: 18H hexaferrites, Acta Mater. 231 (2022) 117854.

2. What is the definition of the effective permeability and how is it related to the antenna structure? Do the slots also have the influence on the effective permittivity?

3. What does the color mapping in Figure 3(c) mean? It should be mentioned in the figure caption and detailed discussion should be included in the text.

4. Why did the author choose 3 mm as the thickness of the MDM in the antenna design? What is the effect of the thickness on the antenna properties?

5. Besides S11 and input impedance, are other antenna parameter, such as gain pattern and efficiency, characterized to evaluate the MDM antenna performance?

6. Are there any performance improvement using the MDM compomer to the original antenna?

7. The written language of the manuscript should be further improved.

Author Response

Report (Reviewer 1)

In this paper, the authors achieved the miniaturization of a 130-MHz planar monopole antenna using a low-loss planar magneto-dielectric material. The new antenna topology was realized by introducing slots partially covered by the planar magneto-dielectric material and the miniaturization rate was about 60%. While the paper is quite straightforward, it could be further improved by the following suggestion or discussion.

We would like to thank the Reviewer for his comments and remarks that helped us to improve the quality of our manuscript. All comments have been taken into account carefully.

  1. In Introduction, the review on the magneto-dielectric materials is insufficient. Discussion on different types of the MDMs and their capabilities on antenna miniaturization should be mentioned. The discussion in the paper shown below is closely related to this topic and is worth citing.
  2. Li, Y. Chen, C. Yu, L, Young, J. Spector, V.G. Harris, Emerging magnetodielectric materials for 5G communications: 18H hexaferrites, Acta Mater. 231 (2022) 117854.

the reference has been added and commented to the introduction (reference [13])

  1. What is the definition of the effective permeability and how is it related to the antenna structure? Do the slots also have the influence on the effective permittivity?

The definition and the calculation formula of effective permeability has been added in the section 4 (lines 113 to 123)

  1. What does the color mapping in Figure 3(c) mean? It should be mentioned in the figure caption and detailed discussion should be included in the text.

It’s the intensity of the surface currents. The caption of figure 3 is modified.

  1. Why did the author choose 3 mm as the thickness of the MDM in the antenna design? What is the effect of the thickness on the antenna properties?

The P-MDM thickness is limited to 3 mm because of technical reasons. the material thickness has a direct effect on the effective medium (effective permeability and permittivity). In our application, the increase of the P-MDM thickness leads to an effective permeability more important. (text added – lines 105 to 110)

  1. Besides S11 and input impedance, are other antenna parameter, such as gain pattern and efficiency, characterized to evaluate the MDM antenna performance?

We have measured the realized gain (free space outdoor measurement facilities not suitable for radiation pattern). It is compared to the simulated results in figure 16. The simulated radiation pattern at 130 MHz has been also added.

  1. Are there any performance improvement using the MDM compomer to the original antenna?

With a very high miniaturization rate such as 60 %, we cannot expect an improvement in any performance. But using MDM the degradation of performance was reduced compared to the other methods of miniaturization. The radiation Q is low enough to show that the antenna can be matched easily with good radiation efficiency.

  1. The written language of the manuscript should be further improved. The English writing has been revised.

Reviewer 2 Report

In this work, the authors proposed a novel method to obtain a miniaturized planar monopole antenna using in 100-200 MHz by placing the slots between two small pieces of the low loss planar magneto-dielectric material (P-MDM). The effects of the position and the dimensions of the P-MDM together with the dimensions of slots on the miniaturization rate were discussed in experimental and simulated. The results show that a higher effective magnetic medium can be obtained by increasing the concentration of the magnetic field intensity, and thus a higher miniaturization rate can be obtained. This research work is informative and logical. I believe the manuscript is suitable for publication in the Magnetism after minor revision.

Following are the comments and the questions:

1.      In the part of “MDM description”, the authors mention that the Ni-Zn-Co ferrite is elaborated and presented in the reference [5]. However, it seem that the electromagnetic parameters of the Ni-Zn-Co ferrites in [5] are different with that of the Ni-Zn-Co ferrites in this paper. I suggest that the component, sintering conditions, and morphology of the Ni-Zn-Co ferrite block need to be displayed in this section.

2.      In the part of “miniaturization of the monopole antenna using a P-MDM” (line 113), the effective permeability is calculated as 2.28 and 4.6 for the antenna A2 together with the antenna A3, respectively. Please state the exact calculation methods or formulas to obtain the effective permeability mr in this section.

3.      In addition to reduce the dimensions of the antenna, the damage to the radiation gain performance should be avoided as much as possible. Therefore, it is necessary to supply the simulated and measured radiation gain results of the matched antenna A5 together with the matched antenna A1 at around the resonance frequency 130 MHz. In this way, the quality of this work could be improved further.

4.      The marker in Fig.3 (a) possibly mislabeled as “6 mm” rather than “60 mm”, please check this point carefully.

5.      Where is the Fig.15 (a) and (b)? Please check this point carefully.

6.      In the lines 88 and 89, the contents “Characteristics of the planar monopole antennas a1 and ARef.” should be removed.

Author Response

 In this work, the authors proposed a novel method to obtain a miniaturized planar monopole antenna using in 100-200 MHz by placing the slots between two small pieces of the low loss planar magneto-dielectric material (P-MDM). The effects of the position and the dimensions of the P-MDM together with the dimensions of slots on the miniaturization rate were discussed in experimental and simulated. The results show that a higher effective magnetic medium can be obtained by increasing the concentration of the magnetic field intensity, and thus a higher miniaturization rate can be obtained. This research work is informative and logical. I believe the manuscript is suitable for publication in the Magnetism after minor revision.

We would like to thank the Reviewer for his comments and remarks that helped us to improve the quality of our manuscript. All comments have been taken into consideration

Following are the comments and the questions:

  1. In the part of “MDM description”, the authors mention that the Ni-Zn-Co ferrite is elaborated and presented in the reference [5]. However, it seem that the electromagnetic parameters of the Ni-Zn-Co ferrites in [5] are different with that of the Ni-Zn-Co ferrites in this paper. I suggest that the component, sintering conditions, and morphology of the Ni-Zn-Co ferrite block need to be displayed in this section.

    the final composition is patented. It cannot be disclosed. The reference [5] is to mention the fabrication method, it’s not the same material that we used (text was modified – line 70)
  2. In the part of “miniaturization of the monopole antenna using a P-MDM” (line 113), the effective permeability is calculated as 2.28 and 4.6 for the antenna A2 together with the antenna A3, respectively. Please state the exact calculation methods or formulas to obtain the effective permeability mr in this section.

 The definition and the calculation formula of effective permeability has been added in the section 4 (lines 110 to 123)

  1. In addition to reduce the dimensions of the antenna, the damage to the radiation gain performance should be avoided as much as possible. Therefore, it is necessary to supply the simulated and measured radiation gain results of the matched antenna A5 together with the matched antenna A1 at around the resonance frequency 130 MHz. In this way, the quality of this work could be improved further.

 We have measured the realized gain. It is compared to the simulated results in figure 16. The radiation pattern wasn’t measured because we are not able to measure the antenna at this frequency (130MHz).

  1. The marker in Fig.3 (a) possibly mislabeled as “6 mm” rather than “60 mm”, please check this point carefully.

The error was corrected

  1. Where is the Fig.15 (a) and (b)? Please check this point carefully.

 Figure 15 is already in page 8 (figure title in lines 220-221)

  1. In the lines 88 and 89, the contents “Characteristics of the planar monopole antennas a1 and ARef.” should be removed.

Table 1 has been removed

Reviewer 3 Report

The authors presented a new topology of the planar monopole antenna using a Magneto-Dielectric material for electrically small wideband applications. It is a well presented manuscript and has good technical soundness. However, this reviewer has following concerns:

1. Please, include more recent works in the literature review of the manuscript and show a clear comparison of the proposed work with other most recent works.

2. Please, explain why a percentage of 60 was chosen as a miniaturization target? Is there any specific reason or it is a random choice?

3. This reviewer is also interested to know why monopole antenna is  preferable for heavy duty mobile applications?

4. The figure and captions should be in the same page, for example, Fig. 13 and captions are not in the same page. Those kind of errors should be fixed.

5. Please, include radiation patterns (far-field) measurements of the proposed prototype. 

Author Response

The authors presented a new topology of the planar monopole antenna using a Magneto-Dielectric material for electrically small wideband applications. It is a well presented manuscript and has good technical soundness. However, this reviewer has following concerns:

We would like to thank the Reviewer for his comments and remarks that helped us to improve the quality of our manuscript.

  1. Please, include more recent works in the literature review of the manuscript and show a clear comparison of the proposed work with other most recent works.

Recent work was added in the reference [13] and mentioned in the introduction

  1. Please, explain why a percentage of 60 was chosen as a miniaturization target? Is there any specific reason or it is a random choice?

It’s not a random choice, it’s the scope statement of our project to reduce the height of the antenna used for aeronautical applications from 50 cm to 20 cm. This gives an objective of a 60 % of size reduction. 

  1. This reviewer is also interested to know why monopole antenna is  preferable for heavy duty mobile applications?

    our project is for an aeronautical application, and the planar monopole antenna is the suitable antenna for VHF band (text added – lines 87 to 89)
  2. The figure and captions should be in the same page, for example, Fig. 13 and captions are not in the same page. That kind of errors should be fixed.

All figures are corrected and checked

  1. Please, include radiation patterns (far-field) measurements of the proposed prototype. 

 We have measured the realized gain (free space outdoor measurement facilities not suitable for radiation pattern). It is compared to the simulated results in figure 16. The simulated radiation pattern at 130 MHz has been also added.

Round 2

Reviewer 1 Report

The authors have addressed all my questions. I have no further comments. 

One typo I noticed is in the caption of Figure 17. The unit of the frequency is missing. 

Author Response

Thank you for this careful review.

One typo I noticed is in the caption of Figure 17. The unit of the frequency is missing. 

The frequency unit is added to the legend of figure 17.

Reviewer 3 Report

The concerns/comments are addressed. No further comments.

Author Response

Thank you again for this review